# Biological Action of Singlet Molecular Oxygen from the Standpoint of Cell Signaling, Injury and Death

**DOI:** 10.3390/molecules28104085

**Published:** 2023-05-14

**Authors:** Junichi Fujii, Yuya Soma, Yumi Matsuda

**Affiliations:** 1Department of Biochemistry and Molecular Biology, Graduate School of Medical Science, Yamagata University, Yamagata 990-9585, Japan; 2Graduate School of Nursing, Yamagata University Faculty of Medicine, Yamagata 990-9585, Japan

**Keywords:** photodynamic therapy, ultraviolet, endoperoxides, lipid peroxidation, apoptosis, ferroptosis

## Abstract

Energy transfer to ground state triplet molecular oxygen results in the generation of singlet molecular oxygen (^1^O_2_), which has potent oxidizing ability. Irradiation of light, notably ultraviolet A, to a photosensitizing molecule results in the generation of ^1^O_2_, which is thought to play a role in causing skin damage and aging. It should also be noted that ^1^O_2_ is a dominant tumoricidal component that is generated during the photodynamic therapy (PDT). While type II photodynamic action generates not only ^1^O_2_ but also other reactive species, endoperoxides release pure ^1^O_2_ upon mild exposure to heat and, hence, are considered to be beneficial compounds for research purposes. Concerning target molecules, ^1^O_2_ preferentially reacts with unsaturated fatty acids to produce lipid peroxidation. Enzymes that contain a reactive cysteine group at the catalytic center are vulnerable to ^1^O_2_ exposure. Guanine base in nucleic acids is also susceptible to oxidative modification, and cells carrying DNA with oxidized guanine units may experience mutations. Since ^1^O_2_ is produced in various physiological reactions in addition to photodynamic reactions, overcoming technical challenges related to its detection and methods used for its generation would allow its potential functions in biological systems to be better understood.

## 1. Introduction

Interactions of oxygen molecules with electrons leaked from enzymatic and non-enzymatic processes produces reactive oxygen species (ROS), such as superoxide (O_2_^−^•) and hydrogen peroxides (H_2_O_2_) [1,2,3]. Hydroxyl radicals (HO•) are the most reactive type of ROS and are likely produced by the reaction of H_2_O_2_ and ferrous iron, via the so called Fenton reaction [4], although there is some debate as to which ROS are the primary products in this reaction [5,6]. Singlet molecular oxygen (^1^O_2_), is a high-energy oxygen species but possesses unique properties that are different from other ROS [7,8,9]. While most ROS are produced by electron transfer reactions, ^1^O_2_ is generated by the transfer of energy to the ground state, triplet molecular oxygen (^3^O_2_), the most abundant oxygen molecule in atmosphere. The type II photodynamic reaction promoted by the presence of photosensitizing molecules is widely employed to generate ^1^O_2_ in biological systems [10,11]. However, the problem with the photodynamic action is that other ROS are also generated as byproducts [12,13]. This methodological issue appears to have hindered progress in research on the biological effects of ^1^O_2_, despite its importance.

^1^O_2_ possesses high energy and is considered to be a major cause for skin damage induced by ultraviolet (UV) irradiation [14,15]. Meanwhile, due to its strong cytotoxicity, ^1^O_2_ is the molecule that is responsible for killing tumor cells during photodynamic therapy (PDT) [10,11]. ^1^O_2_ as well as HO• preferentially reacts with conjugated double bonds, and hence polyunsaturated fatty acids (PUFA), which are dominantly present in the form of phospholipids in the cell membrane, are likely targets [9]. It is known that, upon mitotic stimuli, a small amount of ROS, notably H_2_O_2_, is produced, and this species modulates phosphorylation-mediated signaling pathways [16,17]. While signal modulation by H_2_O_2_ involves the transient oxidation of cysteine (Cys), reactions with ^1^O_2_ tend to result in irreversible oxidation. In most cases, exposure to ^1^O_2_ impairs cellular function but also occasionally stimulates tumorigenic cell growth [18,19]. Concerning cell death, results reported in many studies indicate that the apoptotic pathway is activated by ^1^O_2_ [20,21,22]. However, recent studies suggest that ferroptosis, an iron-dependent necrotic cell death [23,24], is also involved in ^1^O_2_-promoted cell death [25,26].

In this review article, we outline the characteristics and biological actions of ^1^O_2_ and then discuss cell injury and death induced by ^1^O_2_. Because details of the chemical reactions of ^1^O_2_ and its clinical applications to PDT are not the primary focus of this article, readers are referred to corresponding reviews on these issues [8,9,27].

## 2. Fundamental Knowledge of the Biochemical Properties of ^1^O_2_

Because ^1^O_2_ is a major oxidant that is responsible for photoaging and for tumor killing during PDT, numerous studies have been conducted from these perspectives in in vitro systems. It is important to understand its generation and reactions with biological molecules in order to achieve a correct evaluation of the significant roles of ^1^O_2_ in vivo.

### 2.1. Properties of ^1^O_2_ as a Potent Oxidant

Oxidative stress is induced by either the production of large amounts of ROS or an insufficient amount of antioxidants which include enzymes that eliminate ROS or small antioxidant compounds, such as glutathione (GSH), carotenoids and tocopherols [2]. Electrons that are leaked from enzymatic and non-enzymatic reactions initiate the generation of ROS, as represented by O_2_^−^•, H_2_O_2_ and HO•, and hence, the radical electron plays pivotal roles in the development of oxidative stress in many situations [1,3]. However, ^1^O_2_ is generated when the oxygen molecule in the ground triplet state ^3^O_2_ is excited by receiving energy without the transfer of an electron. The ^1^O_2_-generating system involves enzymatic reactions, such as myeloperoxidase, lipoxygenase and cyclooxygenase as well as chemical reactions, such as O_2_^−^•-mediated GSH oxidation and the interaction of peroxides with hypochlorite or peroxynitrite [13,28,29,30]. Photoaging and PDT are the subjects that have been most extensively investigated in terms of ^1^O_2_–mediated reactions that are associated with human physiology and the pathogenesis of related diseases. In spite of the high oxidizing power, reactions of ^1^O_2_ are thought to exert only limited effects compared to those of HO• in biological systems.

HO• is considered to be the most reactive ROS and appears to be responsible for a variety of pathological conditions. However, the half-life of HO• is quite short (10 nsec), so it only reacts with molecules that are in close proximity to the site where it is generated. On the other hand, the half-life of ^1^O_2_ is approximately 4 µsec in aqueous solution, which allows it to diffuse 150–220 nm [31,32]. Thus, ^1^O_2_ may react at various locations beyond where it is generated and, therefore, can affect surrounding molecules and organelles more widely compared to HO•. Nevertheless, this distance is insufficient for extracellularly produced ^1^O_2_ to move to the interior of a cell. Therefore, ^1^O_2_ that is generated inside the cell has the ability to damage various cellular components including DNA and organelles.

### 2.2. Chemical Probes for Detecting ^1^O_2_

Analyses employing a cell biological approach are essential for answering basic questions as to which part of the cell produces ^1^O_2_ in photoaging and during PDT and how cellular responses proceed in such situations. For that purpose, the use of a fluorescent chemical probe is the most convenient approach. ^1^O_2_ sensor green (SOSG) is a prototype that is popularly used in studies for detecting ^1^O_2_, although it has some disadvantages such as lack in membrane permeability [33]. Other compounds have been designed to overcome the disadvantage of SOSG. For example, Aarhus Sensor Green, which is a tetrafluoro-substituted fluorescein derivative that is covalently linked to a 9,10-diphenyl anthracene moiety [34] and the classic indocyanine green probe may also be applicable for this objective in certain experiments [35]. To increase the cellular delivery of SOSG, biocompatible nanosensors, with SOSG encapsulated within their hydrophobic core, have been developed, and these modifications appear to improve its delivery [36,37,38].

The cell membrane permeable far-red fluorescence probe Si-DMA, which is composed of silicon-containing rhodamine and anthracene moieties as a chromophore, has also been developed [39]. Upon reaction with ^1^O_2_, Si-DMA is converted into an endoperoxide at the anthracene moiety that emits strongly. The use of Si-DMA reportedly enables the visualization of ^1^O_2_ generated in a single mitochondrial tubule during PDT. After the treatment of cells with the endoperoxide, dose-dependent increases in fluorescence of Si-DMA were observed [40]. Thus, these results suggest that chemical probes may be applicable for studies concerning the cellular effects of ^1^O_2_. This compound is now commercially available. Another compound, a rhodamine 6G-aminomethylanthracene-linked donor–acceptor molecule (RA), was reported to exhibit unique properties [41]. RA acts as a fluorogenic ^1^O_2_ sensor molecule and also acts as a photosensitizer to generate ^1^O_2_ upon exposure to green light. Other fluorescent reagents, such as one based on an aminocoumarin-methylanthracene-based electron donor–acceptor molecule [42], are also being developed. Since information on the use of these newly developed probes is currently limited, it is necessary to carefully choose which compounds are suitable for the intended research.

### 2.3. Photodynamic Reaction as a ^1^O_2_ -Generating System

PDT mainly contributes to ^1^O_2_ generation in biological systems through type II mechanisms that involve energy transfer from triplet excited molecules to triplet oxygen. Photosensitizers may also act according to competitive type I photosensitization mechanism that mostly involves charge transfer between suitable targets and a photosensitizer in its triplet excited state [43,44]. In order to protect cells against the deteriorating action of UV light, the effects of ^1^O_2_ on skin tissues have been extensively investigated. In the meantime, ^1^O_2_ is considered to be the main molecule that promotes cytotoxic processes during PDT, and hence, multiple studies are currently underway with the aim of understanding the mechanism responsible for ^1^O_2_-mediated cell death and developing efficient photosensitizers [8,10]. UV radiation causes skin photoaging and oxidatively generated damage to dermal cells and is especially troublesome in cases of sunburn which occurs by exposure to excessive UV for long periods of time [14,45]. UVB (280–315 nm) comprises approximately 5% of the solar UV and causes the direct photodamage to many molecules including DNA and proteins in skin tissues through its high energy photochemical reactions. Genetic damage caused by the oxidative modification of DNA and other molecules emerges in a short time after exposure to UVB. In the case of UVA (315–400 nm) that accounts for approximately 95% of the solar UV, cellular damage occurs through the activation of chromophores that act as photosensitizers to generate ^1^O_2_ and other ROS, and hence, the oxidative reaction proceeds indirectly via the ROS that are generated. It is rather difficult to determine if changes in cells that have been exposed to UVA are the consequence of the generation of either ^1^O_2_ or other ROS because they are produced simultaneously by the photodynamic reactions and result in essentially the same end products [46].

In order to elucidate the reactions caused by ^1^O_2_, reliable methods for generating ^1^O are required [12]. The most common method for this purpose is irradiation of the photosensitizer with UV or visible light because it is simple and easy to control its production [27,47]. Figure 1 represents the “Jablonski diagram” that depicts conceptual images of the generation of ^1^O_2_ by the irradiation of a photosensitizing molecule (S) with light followed by transferring energy to ^3^O_2_ [48]. When a photosensitizer is exposed to light, most likely the UVA in natural light, photon energy converts the photosensitizer in the ground state (^0^S) to that in the excited state (^1^S). On returning to the ground state ^0^S, a part of the released photoenergy can be transferred to ^3^O_2_, which results in the electron spin state being altered and the generation of ^1^O_2_. Under this situation, photodynamic action generates not only ^1^O_2_ but also other ROS such as O_2_^−^• and HO• [8].

To observe cellular responses to ^1^O_2_, cell-permeable and non-cytotoxic compounds need to be used as the photosensitizer. For example, Rose Bengal and methylene blue meet the conditions and, hence, are popularly used for the purpose of examining biological action of ^1^O_2_ [47]. Since PDT is a useful therapeutic for eliminating tumors, many attempts have been made to improve the treatment by developing convenient photosensitizers [11,27,49]. This issue, i.e., the applications of photosensitizers, is not discussed further because it is beyond the scope of this review, which is focused on the underlying mechanisms concerning the biological action of ^1^O_2_.

### 2.4. Endoperoxides as Donor Compounds for Generating Pure ^1^O_2_

Endoperoxides that release ^1^O_2_ without other ROS have been developed to evaluate its unique reaction [13,50]. Naphthalene endoperoxide-based ^1^O_2_ donor compounds were first developed, and the structure–function relationships of the compounds have been described in detail in a review article [13]. Consequently, several naphthalene endoperoxides have been established and utilized for the in vitro evaluation of the effects of ^1^O_2_, as representative structures in Figure 2A. Upon mild heating at 37 °C, the endoperoxides spontaneously release pure ^1^O_2_, which then directly reacts with surrounding compounds and organelles. Heat-labile endoperoxides are considered clean sources of ^1^O_2_ for highly specific oxidation of cellular biomolecules and have been applied for ^1^O_2_-mediated oxidation of the DNA guanine base in cells [51,52].

Here we discuss the advantages and disadvantages of ^1^O_2_ donor compounds in comparison with the photodynamic action. The benefits include the following: (1) Endoperoxides produce pure ^1^O_2_. (2) The concentrations of the released ^1^O_2_ are easily controlled. (3) Heating at physiological temperature, generally under cell culture conditions, can promote the release of ^1^O_2_ from endoperoxides. (4) It may be possible to design endoperoxides that are localized to a specific organelle by appropriate chemical modification of the compounds. Limitations include the following: (1) A high concentration of endoperoxides is required to generate sufficient levels of ^1^O_2_. (2) It is essential to consider effects of the raw material after the release of ^1^O_2_ because they are sometimes toxic to cells. (3) Endoperoxides may not be evenly distributed inside cells due to their chemical nature. (4) The amount of ^1^O_2_ released is initially maximal but then gradually decreases with increasing endoperoxide consumption.

When endoperoxides are applied to a cell culture system, it is necessary to use compounds that are able to pass through the cell membrane. In fact, the side chains of naphthalene endoperoxides determine whether they enter cells or remain outside of cells (Figure 2B) [53]. 1-Methylnaphthalene-4-propanonate endoperoxide (MNPO_2_) is cell membrane-permeable and generates ^1^O_2_ within cells. However, 1,4-naphthalenedipropanoate endoperoxide (NDPO_2_) cannot enter cells. Accordingly, while MNPO_2_ induces cell damage, NDPO_2_ at the same concentration has no effects, although both compounds trigger the release of cyt c from isolated mitochondria to a similar extent [53]. We have not examined 1,4-dimethylnaphthalene endoperoxide (DMNO_2_), but it can also enter cells due to the hydrophobic property of its side chains.

While the generation of ^1^O_2_ by a light-irradiated photosensitizer is frequently used, the use of ^1^O_2_ donor compounds has been limited because they are complex molecules that are difficult to synthesize. Some naphthalene endoperoxides are now commercially available. New compounds other than naphthalene-based endoperoxides are also being developed. For the efficient delivery of a ^1^O_2_ donor to cancer cells, a porphyrin-based covalent organic framework that contains a naphthalene endoperoxide has also been developed [54]. Trials to develop new types of endoperoxides, which are based on 2-pyridone and anthracene, are also underway [50]. Two ^1^O_2_-producing systems, photodynamic reactions and naphthalene endoperoxides, have provided rather consistent results so far [13], implying that the contribution of other byproducts may be negligible. Development of these systems will hopefully accelerate our understanding of the nature of ^1^O_2_.

### 2.5. Natural or Synthetic ^1^O_2_-Scavenging Compounds

The body is protected from oxidatively generated damage by a variety of natural and synthetic compounds that scavenge ^1^O_2_. The quantitative evaluation of the ^1^O_2_-scavenging ability of a compound provides useful information not only for basic research but also for developing functional foods and medicines concerning antioxidation [55]. Many nutritional compounds, such as tocopherols, carotenoids and flavonoids, possess antioxidant capacity and protect susceptible molecules from ^1^O_2_. The oxygen radical absorption capacity (ORAC) assay is a representative method for the detection of ^1^O_2_-scavenging ability of food ingredients [56]. Thereafter, a simple method called a singlet oxygen absorption capacity (SOAC) assay has been established for the evaluation of ^1^O_2_-scavenging ability [56,57]. These methods are useful in exploring popularly used ^1^O_2_-scavenging compounds, especially in the field of food chemistry.

While carotenoids react with ^1^O_2_ more rapidly than α-tocopherol, in a nearly diffusion-limited manner (~10^10^ M^−1^s^−1^) [58,59], lycopene, which is found in fruits and vegetables such as tomato, is one of the strongest natural carotenoids [60]. By employing the SOAC assay method, carotenoids have been found to quench ^1^O_2_ approximately 30–100 times faster than α-tocopherol [57]. After transferring excitation energy to carotenoids, ^1^O_2_ returns to the ground state. The excited carotenoids spontaneously release thermal energy and then return to the ground state. Hence, carotenoids are spontaneously recycled and have the advantage of quenching ^1^O_2_ without affecting other molecules. This chemical property of ^1^O_2_ is a major difference from other radicals that are generated by electron transfer reactions that require another radical to quench.

Based on in vitro data on the action of anti-oxidants, the biological benefit of carotenoids has also been examined by some studies on humans. The administration of a representative carotenoid β-carotene to humans failed to alleviate sunburn reactions under the conditions used [61]. However, a later study reported that carotenoids, β-carotene and lycopene effectively protect erythema formation induced using a solar light simulator [62,63]. Lycopene has also attracted attention as a nutrient with anticancer effects [64]. For another example, lutein is a xanthophyll carotenoid that is found in foods such as dark green leafy vegetables and exhibits strong antioxidant activity via its ability to scavenge ROS including ^1^O_2_ and lipid peroxyl radicals [65]. Lutein appears to exert anti-inflammatory actions against some diseases, including neurodegenerative disorders, eye diseases, cardiovascular diseases and skin diseases.

Many synthetic antioxidant compounds that scavenge ROS including ^1^O_2_ have been developed as medicines. Edaravone (3-methyl-1-phenyl-2-pyrazolin-5-one) is a compound that eliminates a variety of radical species and was the first approved compound for use as a medicine for the treatment of acute brain infarctions. Edaravone can scavenge ^1^O_2_ that is generated by activated human neutrophils [66] and by photoactivated Rose Bengal [67]. The plasma lipid peroxidation caused by ^1^O_2_, however, cannot be suppressed by edaravone and other clinical drugs with antioxidant ability, which include roglitazone, probucol, carvedilol, pentoxifylline and ebselen, although they exhibit suppressive effects on lipid peroxidation caused by free radicals, peroxynitrite, hypochlorite, and lipoxygenase reactions [68]. Because blood plasma contains high concentrations of proteins and many other compounds that could potentially interfere with the scavenging reaction by these chemicals, such biological compounds may have influenced the results.

## 3. Oxidative Modification of Biological Molecules and Damage to Organelles by ^1^O_2_

^1^O_2_ preferentially reacts with double bonds but can also react with sulfhydryl groups of Cys in proteins. In fact, ^1^O_2_ oxidizes important biological molecules including lipids, proteins and nucleic acids, while oxidation end products by ^1^O_2_ are largely similar to those produced by other ROS.

### 3.1. Lipid Peroxidation by ^1^O_2_

Both ^1^O_2_ and HO• effectively oxidize lipids, leading to the formation of lipid hydroperoxides (LOOH), although the pathways and efficiency of their production differ between these oxidants [2,8]. Because biological membranes contain large quantities of phospholipids and cholesterol, their oxidative modification may cause the destruction of the cell membrane. Mono-unsaturated fatty acids (MUFA) and poly-unsaturated fatty acids (PUFA) are both targets of ^1^O_2_, but PUFA are among the preferred target that is present in the cell membrane as well as lipid droplets in the form of triacylglycerol molecules. ^1^O_2_ also oxidizes cholesterol, which results in the formation of several oxidized products that are collectively referred to as oxysterols [69,70,71]. Accordingly, cholesterol hydroperoxides are specific biomarkers of type I and type II photosensitization reactions [72,73]. Phospholipids generally contain unsaturated fatty acids at the sn-2 position, which may undergo oxidation by ^1^O_2_ (Figure 3A). In the presence of metal ions or HOCl, P-LOOH may also generate ^1^O_2_ based on a Russell mechanism [74]. Because the resultant oxidation products are structurally the same irrespective of the causal oxidant, ^1^O_2_ and HO•, it is difficult to determine which reactive species actually caused the lipid peroxidation.

Following the peroxidation of lipids, the β-scission of LOOH may release small aldehyde molecules, as represented by 4-hydroxyl nonenal, which impairs cellular function but may also act as a signaling molecule [46,75,76]. The accumulation of P-LOOH along with iron may consequently induce cell rupture [77], which is typically observed in ferroptosis [23,24]. Although the accumulation of lipid-derived aldehydes, such as 4-hydroxyl nonenal and malondialdehyde, is considered to be a hallmark of ferroptosis [78,79], it is not clear at this moment if the β-scission of P-LOOH is essential or not for executing cell death. LOOH produced by ^1^O_2_ during PDT along with iron may produce lipid radicals LOO• and then initiate a chain reaction, which leads to cell death [80]. Nitric oxide reacts with LOO• to form LOONO, which terminates the chain reaction and hence suppresses cell death [81]. Anti-cancer chemotherapy generally induces cell death by means of the production of ROS, while some malignant tumors overexpress nitric oxide synthase (NOS). These collective observations suggest that the inhibition of nitric oxide synthesis would increase the efficacy of killing the NOS-expressing tumors during PDT [82].

### 3.2. Oxidation of Amino Acids and Proteins by ^1^O_2_

Proteins are present in high concentrations in the cytosol and organelle membranes. Side chains of amino acids in proteins contain reactive groups which reportedly consume approximately 68% of the ^1^O_2_ that is generated inside cells. ^1^O_2_ selectively reacts with only a few amino acids in proteins, which include Cys, methionine (Met), histidine (His), tyrosine (Tyr) and tryptophan (Trp) [8]. When cyt c is employed as a model protein, Met, His and Trp were reported to be preferentially oxidized by the ^1^O_2_ generated by the photoexcitation of phthalocyanine Pc 4 [83]. Interestingly, the same amino acids in lysozyme are oxidized by the ^1^O_2_ released from endoperoxides [84]. The oxidation of the same amino acids, despite the use of different ^1^O_2_ sources and different model proteins, suggests that these amino acids are also susceptible targets of ^1^O_2_ in other proteins as well.

While sulfhydryl groups in isolated proteins are largely present in the form of disulfide bonds, this makes them resistant to further oxygenation. Intracellular proteins mainly contain Cys in the reduced form (Cys-SH) due to the reducing environment inside cells. Oxidation converts Cys and GSH to the oxidized Cys dimer (cystine) and oxidized glutathione dimer (GSSG), respectively. The oxidation of sulfhydryl groups in proteins yields three additional forms of oxidation products, depending on the redox microenvironment [85]. In the absence of nearby Cys or GSH, Cys-SH is oxidized to cysteine sulfenic acid (Cys-SOH), cysteine sulfinic acid (Cys-SO_2_H) or cysteine sulfonic acid (Cys-SO_3_). While Cys-SOH can be reduced back to Cys-SH by means of reductants such as GSH, Cys-SO_2_H and Cys-SO_3_H cannot be reduced to Cys-SH in mammalian cells. Accordingly, enzymes with reactive Cys residues in the catalytic center are permanently inactive after hyperoxidation to Cys-O_2/3_H.

Lysosomes are organelles that hydrolyze polymeric compounds that are incorporated via endocytosis and those encapsulated within autolysosomes during autophagy. Cathepsins are a series of lysosomal acidic proteases and are classified into three groups, based on their catalytic amino acids: serine proteases, cysteine proteases and aspartate proteases. Cathepsins B and L/S belong to the cysteine protease group and are abundantly present in many mammalian tissues [86], while cathepsins D/E are representatives of the aspartate protease group [87]. The release of ^1^O_2_ from a naphthalene endoperoxide selectively inactivates isolated cysteine proteases cathepsins B and L/S but not aspartate proteases cathepsins D/E in vitro [88]. Serine proteases, trypsin and chymotrypsin, as well as other cysteine proteases, such as papain and calpain II, have also been reported to be inactivated by ^1^O_2_ [89]. The hydroxyl group in the side chain of serine (Ser) is generally less reactive compared to Cys-SH, but the tertiary structure of the catalytic center of a serine protease, named the serine-histidine-aspartate catalytic triad, makes it highly reactive, which enables electrophilic attack at the peptide bond of the substrate protein. As a result of these properties, the hydroxyl group of the catalytic Ser is highly susceptible to oxidative insult, as has also been found for the catalytic Cys in cysteine proteases. This mechanism may explain the high susceptibility of trypsin and chymotrypsin to ^1^O_2_ and results in a high sensitivity comparable to that for cysteine proteases [89].

Phosphotyrosine phosphatases (PTP) are another enzyme family that possesses reactive Cys for catalysis, as found in cysteine proteases. The high sensitivity to oxidants of the catalytic Cys in PTP allows modulation of the phosphorylation signal by hydrogen peroxide [16,17]. ^1^O_2_ from the endoperoxide oxidatively inactivates protein tyrosine phosphatase-1B (PTP1B), which plays a pivotal role in cellular signal transduction and undergoes redox regulation [90]. A mass-spectrometric analysis identified Cys-OH and Cys-O_2_H as oxidized amino acids in ^1^O_2_-treated PTP1B [91]. Thus, these Cys-centered enzymes appear to be common targets of ^1^O_2_ (Figure 3B). This issue is discussed further below in explaining the biological action of ^1^O_2_.

### 3.3. Oxidation of Nucleotides by ^1^O_2_

UVB can lead to the development of skin cancer. The incidence of skin cancer is greatly increased in xeroderma pigmentosum (XP) patients who have defects in the nucleotide excision repair system [92]. However UVB itself is considered to be a very poor DNA oxidizing agent in most living systems. ^1^O_2_ causes the oxidative modification of nucleotides [93,94]. 2′-Deoxyguanosine is the most reactive nucleoside compared to other nucleosides and produces 8-oxo-7,8-dihydro-2′-deoxyguanosine (8-oxodG) (Figure 3C). The rate of reaction of ^1^O_2_ with 2′-deoxyguanosine is approximately two orders of magnitude faster than that with other nucleosides [8]. This trend is similar to that for the oxidative modification of 2′-deoxyguanosine by HO•, which also produces 8-oxodG [95]. The presence of an 8-oxodG residue in DNA frequently causes a G to T transversion, which is associated with mutations of genes and may lead to the development of tumors. UVB radiation dominantly causes dimeric pyrimidine photoproducts, while 8-oxoG represents at best 1% of the overall amount of dimeric pyrimidine photoproducts generated by exposure to UVB radiation [96,97]. Accordingly, UVB-induced skin cancer is essentially due to the photo-induced formation of cytosine-containing bipyrimidine photoproducts but not 8-oxoG formation. Meantime UVA generates ^1^O_2_ with respect to HO•-mediated oxidation as well as cyclobutane pyrimidine dimers as the main photoproducts. Thus, UVA is considered to be a better DNA-oxidizing agent [98]. These collective findings imply that the excessive exposure to UV, not only to UVB but also to UVA, may increase the incidence of skin cancer [99]. ^1^O_2_ also oxidizes NAD(P)H to NAD(P)^+^, which appears to be associated with dysfunctional mitochondrial metabolism [100].

While intact DNA is not antigenic in the general sense, that treated with ^1^O_2_ exhibits a high immunogenicity and may become an epitope for autoantibodies in autoimmune diseases such as systemic lupus erythematosus (SLE) [101]. Consistent with this mechanism, a neoepitope is, in fact, generated by ^1^O_2_ exposure and can be recognized by immunoglobulin G [92,102]. Exposure to UVA in the presence of excessive endogenous photosensitizing molecules, such as porphyrin, increases the generation of ^1^O_2_, which may also increase the incidence of skin cancer [94]. Porphyrin-related compounds accumulate in porphyria that are divided into eight distinct groups by the defect in the genes involved in the synthesis of heme. Accumulated porphyrin-related compounds act as photosensitizers upon exposure to sunlight. Generated ROS including ^1^O_2_ are considered to be largely involved in the hypersensitivity of skin in porphyria, while protein aggregation induced by direct interactions with porphyrin-related compounds appears to play roles in pathogenic damage in internal organs such as the liver [103].

### 3.4. Organelle Damage by ^1^O_2_

Concerning oxidatively generated damage in cells, some organelles are also sensitive targets of ^1^O_2_, and mitochondria and lysosomes are the primary target organelles of ^1^O_2_ generated by PDT [104]. A sublethal dose of UVA irradiation causes the common deletion in mitochondrial DNA in human fibroblasts, which can be attributable to the generation of ^1^O_2_ within mitochondria [105]. Mitochondrial permeability transition is either activated or inactivated by ^1^O_2_, depending on the site where the ^1^O_2_ is produced [106], whereas Bcl-2 may also be a sensitive target of ^1^O_2_ and responsible for apoptosis [107]. Irradiation with a 1267 nm laser produces ^1^O_2_ without other ROS and induces apoptosis by opening mitochondrial permeability transition in B16 melanoma cells but has no effects on fibroblasts [22]. In the case of B16 melanomas, melanin and its precursor molecules, which act as both photoprotectors and photosensitizers [108], are abundantly produced and, upon laser irradiation, may exert an influence on the sensitivity of the cells.

^1^O_2_ largely oxidizes mitochondrial membranes, leading to the stimulation of autophagic degradation of damaged mitochondria, a process that is referred to as mitophagy [109,110,111]. Moreover, lysosomes are also one of sensitive targets of PDT [112]. The destruction of lysosomes by oxidatively generated damage during PDT releases a variety of hydrolytic enzymes that may cleave cytosolic components including organelle proteins and lipids. Because lysosomes store relatively large amounts of iron, which is originated from transferrin through endosomes [113], the stored iron is also released during PDT. Because iron is involved in the production of oxygen radical species including HO•, its release accelerates cell dysfunction and death [6]. Thus, the destruction of lysosomes along with mitochondrial damage coordinately stimulates cytotoxic effects and enhances the therapeutic efficiency of PDT [114,115]. Based on this mechanism, photosensitizers that target mitochondria or lysosomes are considered to be more effective in tumoricidal action [116,117].

## 4. Cellular Responses to ^1^O_2_

The high reactivity of ^1^O_2_ is not only a tumoricidal component of PDT but is also applicable for sterilizing microorganisms [118,119,120]. The generation of excess levels of ^1^O_2_ causes cell damage and death [112,121], whereas physiological levels of ^1^O_2_ likely act as modulators of cellular signaling, as typically seen in the action of hydrogen peroxide [16].

### 4.1. O_2_ in Protection against Microbial Infection

ROS and reactive nitric oxide species (RNOS) are recruited by the immune system to combat microbial infections. Neutrophil extracellular traps (NET) play pivotal roles in innate host defense by enveloping invading microbes. ^1^O_2_ is produced upon the stimulation of neutrophils with phorbol myristate acetate and is essential for NET formation, independent of ROS production by activated NADPH-oxidase [122]. Meanwhile, ^1^O_2_ generated by either the enzymatic system or from naphthalene endoperoxides directly kills bacteria [118,123]. Accordingly, the antibacterial ability of ^1^O_2_ generated by the photodynamic action has attracted much attention for the purpose of killing multidrug-resistant pathogenic bacteria [10].

Anti-viral action is another beneficial reaction of ^1^O_2_. ^1^O_2_ inhibits infections without damaging viral DNA under the conditions used [100] and may be useful in exterminating viruses in blood components [124]. Efforts are being made to develop photosensitizers that exhibit antiviral actions more effectively by targeting the envelope lipids of viruses [120]. For the purpose of anti-microbial infections, photosensitizers that are activatable by near-infrared light, which penetrates deep into tissue, are being developed and have become another approach to the use of ^1^O_2_ [125].

### 4.2. Redox Modulation of Mitotic Signaling by Oxidants

The role of ROS in mitotic signal transduction, notably in mitotic exit by means of phosphatases, has been elucidated over the past three decades [16,17,126] as shown in Figure 4. Regarding ROS signaling in growth stimuli, a pioneering study on the regulation of PTP1B was performed on cells that had been stimulated with EGF growth factor [127]. In order to understand the potential mechanism for signal regulation by ^1^O_2_, we first described the signal transduction system for growth stimulation followed by its modulation by H_2_O_2_, a major oxidant with moderate oxidizing properties.

Growth factors bind to the corresponding plasma membrane receptor for tyrosine kinase (RTK) at the cell surface and transmit mitotic signals to genes in the nucleus through the phosphorylation of signaling molecules [17,128]. Two types of phosphorylation can occur in amino acid side chains of proteins: Ser/threonine (Thr) and Tyr, whereas the phosphatidylinositide 3-kinase (PI3K)-Akt pathway also involves the phosphorylation of phosphatidylinositol. The binding of the growth factor at extracellular space initiates a signal cascade that proceeds via the autophosphorylation of specific Tyr residues in the cytosolic domain of RTK [128]. The binding of adaptor proteins to phosphotyrosine converts GDP-bound Ras to GTP-bound Ras, allowing the activation of Raf, the most upstream protein kinase in the signaling pathway in mitotic signaling. In the meantime, PI3K is also activated, which results in the stimulation of the protein kinase B/Akt pathway in the case of certain mitogens [11]. The expressions of a series of genes are ultimately stimulated and promote the cell cycle for growth.

The coordinated action of phosphorylation and dephosphorylation reactions of signaling molecules functions to control cell growth properly. While activation of the signaling cascade is corroborated by phosphorylation, dephosphorylation catalyzed by a phosphatase is needed to terminate the signaling to avoid excessive proliferation [129]. Most protein phosphatases are either specific to phospho-Ser/-Thr, designated as PPT, or phospho-Tyr, designated as PTP, while phosphatase and tensin homologs (PTEN) are specific to phosphatidylinositides [90,126,130]. The reaction mechanisms for PPT and PTP during dephosphorylation are completely different, while there is a similarity between PTP and PTEN in that they both employ Cys as the catalytic amino acid.

Upon the activation of RTK, NADPH-oxidase embedded in plasma membranes is activated and produces O_2_^−^•/H_2_O_2_ via a one- or two-electron donation to molecular oxygen on the cell surface [17]. Although O_2_^−^• cannot permeate the plasma membrane due to its negative charge, the presence of the Cl^−^ channel-3 (CIC-3) may allow it in some types of cells [131]. H_2_O_2_ is generally formed either directly by NADPH-oxidase or by superoxide dismutation outside cells after which it then enters the cells [16]. H_2_O_2_ reversibly oxidizes catalytic Cys-SH to Cys-OH, which enables the transient suppression of enzymatic activities of reactive PTP and PTEN upon growth factor stimuli [132,133]. PTP families that include PTP1B, Cdc25, SHP1, and SHP2 have been found to be sensitive targets for controlling this type of ROS signal. While the pKa value of the sulfhydryl of free Cys-SH or the conventional Cys-SH in proteins is approximately 8.5, the pKa value of Cys-SH in the catalytic center of PTP is generally maintained at less than 5.0 due to the microenvironment of the catalytic center. The resulting cysteine thiolate anion (Cys–S^−^) is highly susceptible to oxidation, and even H_2_O_2_ can oxidize Cys–S^−^ to Cys–SOH and inactivate PTP. When H_2_O_2_ is eliminated by the action of peroxidases, such as peroxireodxins, Cys–SOH is reduced back to Cys–S^−^ by redox molecules such as GSH. PTEN also has a catalytic Cys-SH and appears to be regulated by a similar mechanism [117,127]. However, there is concern that extracellularly produced H_2_O_2_ could diffuse into the extracellular space, and that not all of the H_2_O_2_ moves inside the cell. The disadvantage of extracellularly produced O_2_^−^•/H_2_O_2_ may be overcome by forming redoxosomes, which are endosomes that are constructed during RTK internalization [134]. The generation of hydrogen peroxide in the spherical redoxosome allows more hydrogen peroxide to reach the target PTP at the cytoplasmic side.

Concerning the action of ^1^O_2_, both the stimulation [18,19] and the inhibition [135] of cell growth have been reported. Mitochondrial respiration and ATP production in brain cells is stimulated by the ^1^O_2_ generated by a 1267 nm laser pulse [136]. ^1^O_2_ activates Akt via the PI3K pathway, which contributes to a survival response after UVA irradiation [18]. UVA also reportedly activates p38 MAPK in dermal fibroblasts through the activation of ligand-receptor signaling pathways or a ribotoxic stress mechanism [19]. This mitotic stimulation may be associated with the tumor growth activity of ^1^O_2_. In the meantime, however, ^1^O_2_ rapidly disrupts EGF receptor (EGFR)-mediated signaling by decreasing both the protein level and the level of phosphorylation of EGFR [137]. Because the inhibition of protein phosphatases with okadaic acid blocked the dephosphorylation of EGFR, ERK1/2 and Akt, responsible phosphatases appear to be activated by ^1^O_2_ under this condition. Thus, the action of ^1^O_2_ regarding this issue is complicated, and further studies will clearly be needed to elucidate the inconsistent observations concerning the action of ^1^O_2_ on mitotic signaling.

### 4.3. Apoptotic Cell Death Induced by ^1^O_2_

Although several pathways are involved in cell death, in this review, we mainly discuss this issue as it is related to apoptosis and ferroptosis. While apoptosis and necrosis are major classifications of cell death, they are now divided into several subclasses based on morphology and pathways [138]. Apoptosis is the most common type of cell death and is thought to account for more than 90% of physiologic cell death [139]. Apoptotic pathways can be roughly divided into two categories: an extrinsic pathway activated by death receptors and intrinsic pathways involving mitochondria. The extrinsic pathway involves death receptors, notably the Fas and TNFα receptors, and, upon stimulation, procaspase 8 is converted into active caspase 8, resulting in the activation of caspase 3 [140]. In the case of the intrinsic pathway, mitochondria are the first target of apoptotic stimuli, and cyt c is usually then released from mitochondria to the cytosol. While cyt c is a component in the electron transport system that is localized at the mitochondrial inner membrane and transfers an electron from ETC-III to ETC-IV (cytochrome c oxidase), it is released to the cytosol and also plays a key role in promoting apoptosis. The oxidized form of cyt c triggers the assembly of apoptosome that contains Apaf-1 and procaspase 9 as protein components [141,142] and leads to the activation of caspase 9 by autolysis. Procaspase 3 undergoes proteolytic activation by cascade 9, and the resulting active caspase 3 cleaves a variety of proteins, including an inhibitor of caspase-dependent DNase (ICAD). Thus, hydrolytic degradation occurs in proteins and DNA, but membrane structures consisting of phospholipid bilayer remain intact. Accordingly, cellular and nuclear body compaction occurs, but cellular components do not leak out of the apoptotic body in typical apoptosis. Under these conditions, the immune system virtually does not respond to apoptotic cells.

Studies employing photodynamic reactions and endoperoxides imply that apoptotic cell death is induced by ^1^O_2_, as judged by hallmarks including caspase activation and DNA cleavage [143,144,145]. While mitogen-activated protein kinases (MAPK) are originally named based on their roles in mediating mitotic signals [146], ^1^O_2_ and UV irradiation reportedly activates MAPK, p38 MAPK, c-Jun N-terminal kinase (JNK) and extracellular-signal regulated kinase (ERK) members, which eventually leads to the execution of apoptosis [13,147,148]. Because catalytic Cys in PTP is susceptible to ^1^O_2_, the inhibition of PTP may allow the phosphorylated Tyr in the upstream protein kinase to be sustained, leading to the preservation of the active state of downstream MAP kinases [21]. Although it is rather difficult to actually identify the ROS that are involved, one report suggested the differential involvement of ^1^O_2_ and O_2_^−^• in apoptosis pathways by means of cyclosporine A sensitivity to UVA1 (340–400 nm) irradiation [149]. In addition, ^1^O_2_ as well as H_2_O_2_ trigger the cleavage of Bid, which then releases cyt c from mitochondria [150]. This appears to be another mechanism of ^1^O_2_-induced apoptosis, which is independent from the action of the phosphatase/kinase system.

### 4.4. Ferroptosis Can Be an Alternate Cell Death Pathway by ^1^O_2_

About a decade ago, ferroptosis, which is exclusively caused by lipid peroxidation products, was characterized as an iron-dependent necrotic cell death [23,24]. Lipid radicals generated by the presence of ferrous iron are thought to enhance lipid peroxidation reactions and lead to membrane destruction [151]. Ferrous iron donates an electron to H_2_O_2_ which results in the formation of HO• by the Fenton reaction. The resulting HO• then reacts with polyunsaturated fatty acids to produce lipid peroxyl radicals, which initiate radical chain reaction and amplify the production of lipid peroxides [80]. Glutathione peroxidase (GPX4), a major suppressor of ferroptosis, reduces lipid peroxides to alcohols by transferring two electrons from GSH [152]. Different from apoptosis, ferroptosis stimulates the release of intracellular biological molecules, which may be responsible for aggravating inflammation or autoimmune responses under conditions of ^1^O_2_ production [153]. Inflammatory responses to necrotic cell death appear to be consistent with pathological conditions and damage caused by UV irradiation.

Despite the fact that many studies have appeared regarding apoptosis, our study in which MNPO_2_ was used as the ^1^O_2_ donor showed neither the activation of caspases nor DNA ladder formation during the cell death [53]. On the contrary, we found that ^1^O_2_ suppresses etoposide-induced apoptosis, although cell damage is further aggravated. ^1^O_2_ induces the release of cyt c, but it neither causes caspase activation nor the formation of apoptotic bodies. Cell death is alleviated in cells that overexpress GPX4 in mitochondria but not in cytoplasm. Treatment of the oxidized cyt c with MNPO_2_, in fact, completely inhibited its ability to induce caspase activation in an in vitro reconstitution assay. Because treatment with ^1^O_2_ resulted in the oxidation of Met, His and Trp in isolated cyt c [83], it had no effect on the oxidative state of heme [142]. These collective data imply that the oxidation of these amino acids in cyt c by ^1^O_2_ may counteract the induction of apoptosis through disabling interactions with Apaf-1.

Hela cells that had been treated with MNPO_2_ showed elevated hallmarks of ferroptosis, e.g., elevation in lipid peroxidation and free ferrous iron [25]. Ferrostatin-1, which specifically inhibits lipid peroxidation, partly suppresses cell death. The MNPO_2_ treatment had no effect on the status of Cys-GSH [25]. However, glutathione peroxidase and thioredoxin reductase, which are both selenocysteine-containing enzymes, have been reported to be prone to oxidation by ^1^O_2_ generated by photodynamic activation [154,155]. Several studies also have indicated that ^1^O_2_-induced lipid peroxidation is the cause of cell death, implying that ferroptosis is an alternate form of cell death by ^1^O_2_ [26,156].

Accordingly, it would be reasonable to assume that a mixture of apoptosis and ferroptosis is induced by ^1^O_2_ generated during PDT, as was implied [157]. The occurrence of this mixed type of cell death may be attributed to the properties of ^1^O_2_ as schematically depicted in Figure 5. Cyt c is anchored on the inner mitochondrial membrane via an interaction with cardiolipin, which contains four fatty acids, most of which are PUFA. Upon apoptotic stimuli, cyt c is released from mitochondria to the cytosol upon the preferable oxidation of cardiolipin [158]. The oxidized cyt c then assembles with other components to form apoptosomes, which activate caspases, thus resulting in the execution of apoptosis. It therefore appears that cells exposed to moderate levels of ^1^O_2_ may undergo apoptosis (Figure 5A).

However, when exposed to excess ^1^O_2_ (Figure 5B), cyt c and caspase, as cysteine proteases, undergo oxidative inactivation by ^1^O_2_, which aborts the apoptotic pathway [53,142]. Instead of apoptosis, the resulting LOOH and iron released from lysosomes may collectively activate the ferroptotic pathway. Thus, it is conceivable that the dose of ^1^O_2_ may determine if cells die by apoptosis or ferroptosis [22]. This hypothetical mechanism concerning ^1^O_2_-triggered cell death needs confirmation by further experimentation.

## 5. Concluding Remarks

^1^O_2_ is largely produced by photodynamic reactions and, hence, is a major factor in the photoaging of skin, while ^1^O_2_ generated during PDT is major component that exerts tumoricidal activity. Target molecules of ^1^O_2_ are somewhat limited compared to those of HO•, but oxidatively generated damage to multiple molecules and organelles can still occur. Although ^1^O_2_ is produced by many enzymatic and non-enzymatic reactions, the functional significance of generated ^1^O_2_ is poorly understood compared to other ROS. The application of donor compounds endoperoxides as well as newly developed sensor probes could contribute to our understanding of the action of ^1^O_2_ on cellular functions. Since there are contradictory observations on the action of ^1^O_2_ regarding cell growth and death, it will be necessary to clarify these issues.

## Figures and Tables

**Figure 1 molecules-28-04085-f001:**
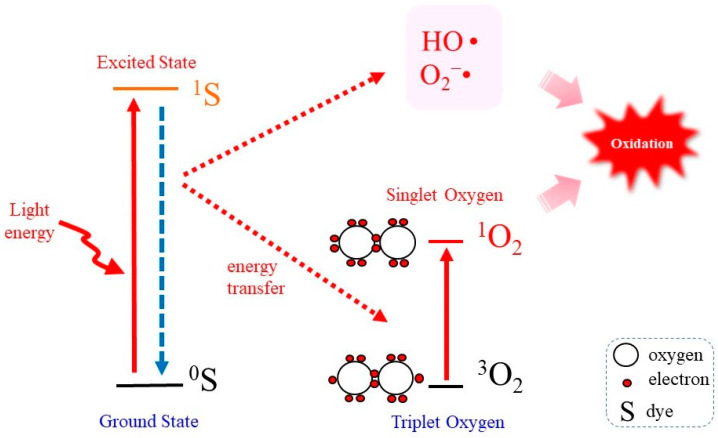
Photodynamic ^1^O_2_ generation. Photoirradiation of a photosensitizing molecule in the ground state (^0^S) leads the production of the excited form (^1^S). On returning to the ground state, energy is transferred to ^3^O_2_, which becomes excited to ^1^O_2_. In the meantime, however, other ROS such as O_2_^−^• and HO• may also be produced.

**Figure 2 molecules-28-04085-f002:**
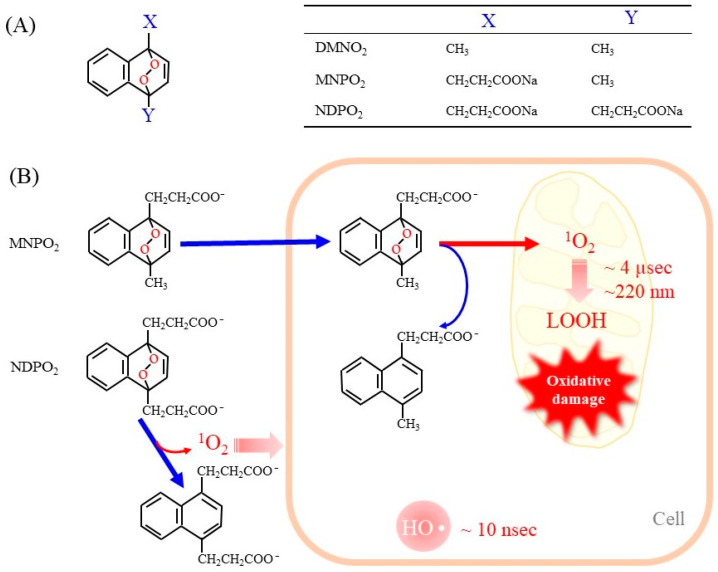
Representative endoperoxides and the release of ^1^O_2_. (**A**) The general structure of naphthalene endoperoxides is shown on the left. The table on the right shows the chemical groups attached to the naphthalene ring. DMNO_2_, 1,4-dimethylnaphthalene endoperoxide; MNPO_2_, 1-methylnaphthalene-4-propanonate endoperoxide_;_ NDPO_2_, 1,4-naphthalenedipropanoate endoperoxide. (**B**) Endoperoxides represented by MNPE and NDPE generate ^1^O_2_ spontaneously at physiologic temperature 37 °C. MNPE, which is relatively hydrophobic, can enter the cell, but NDPE cannot cross the cell membrane. The short life of ^1^O_2_ (~4 µsec) makes it diffuse only 150–220 nm in aqueous solution. As a result, ^1^O_2_ released from NDPE is present outside the cell only, while ^1^O_2_ from MNPE can act inside the cell. For reference, hydroxy radicals are also shown.

**Figure 3 molecules-28-04085-f003:**
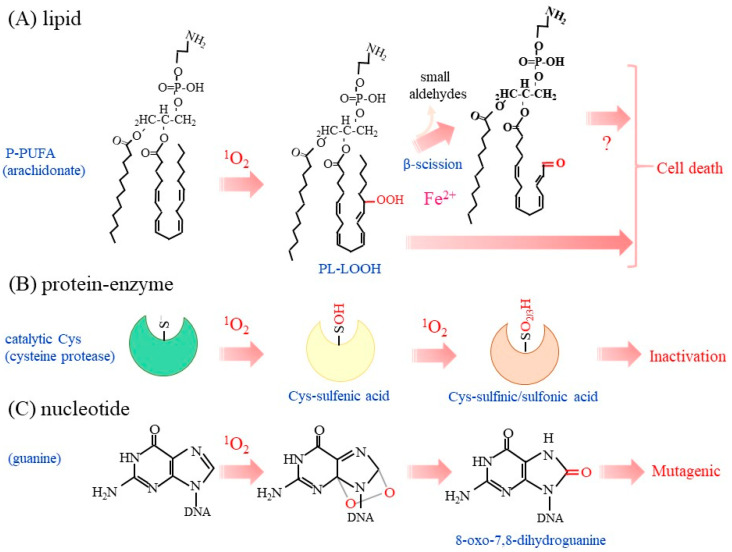
Oxidation of biological molecules by ^1^O_2._ Typical oxidation reaction of a phospholipid, enzymes, and nucleic acids with ^1^O_2_ are depicted. (**A**) Lipid peroxidation preferentially occurs in PUFA in phospholipids, as typified by arachidonic acid. The presence of metal ions, as exemplified by iron, may initiate radical chain reactions and cause the β-scission of LOOH. Cell death may be induced by the accumulated P-LOOH, although it is not clear whether or not β-scission is essential. (**B**) The sulfhydryl group in the catalytic Cys residue of enzymes is most sensitive to ^1^O_2_–induced oxidation in proteins. The Cys-OH may be hyperoxidized to Cys-O_2_H and then Cys-O_3_H, which are collectively presented as Cys-O_2/3_H. (**C**) Purine bases, notably guanine, are prone to oxidation by ^1^O_2_. 8-oxo-7,8-dihydroguanine is consequently produced and may induce mutation in DNA.

**Figure 4 molecules-28-04085-f004:**
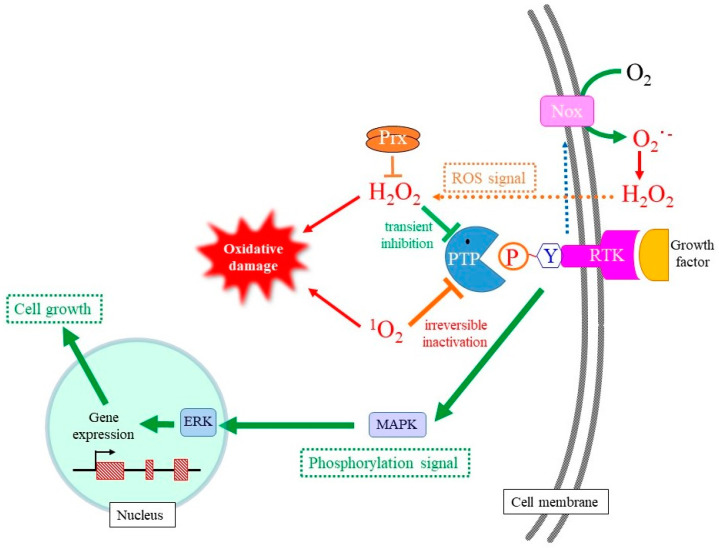
Stimulation of cell growth by hydrogen peroxide and ^1^O_2_. Activation of RTK stimulates the production of O_2_^−^• by NADPH-oxidation and the subsequent formation of H_2_O_2_ via the spontaneous dismutation or an SOD-catalyzed reaction. Because hydrogen peroxide transiently inhibits PTP via the oxidation of catalytic Cys-SH to Cys-SOH, mitotic signaling mediated by Tyr phosphorylation is transmitted to a following signaling molecule. As result, the transient production of hydrogen peroxide upon mitotic stimuli promotes cell growth. ^1^O_2_ also oxidatively modifies the catalytic Cys-SH in PTP, but overoxidation converts it to Cys-O_2/3_H, which cannot be reduced back to Cys-SH. Thus, ^1^O_2_ likely induces the permanent inactivation of PTP, which makes mitotic signaling uncontrollable and leads to tumorigenic cell growth.

**Figure 5 molecules-28-04085-f005:**
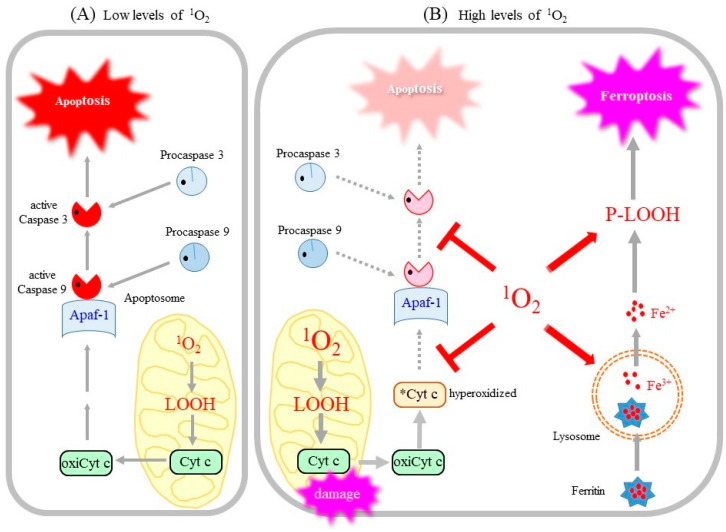
^1^O_2_ dose differentially acts on cells and induces either apoptosis or ferroptosis. (**A**) A low dose of ^1^O_2_ peroxidizes PUFA, notably cardiolipin, results in the release of cyt c from mitochondria. Oxidized cyt c initiates the assembly of apoptosome and results in the activation of the caspase cascade, which leads to the eventual execution of apoptosis. (**B**) A high dose of ^1^O_2_ inactivates cyt c and caspase and aborts the apoptotic pathway. In the meantime, ^1^O_2_ causes lipid peroxidation, which destroys the lysosomal membrane and releases iron. Free iron along with P-LOOH accelerates membrane destruction and leads to the execution of ferroptosis. Mitochondrial damage and the accumulation of P-LOOH are also hallmarks of ferroptosis. oxiCyt c, oxidized, apoptosis-competent cyt c; *Cyt c, oxidized, apoptosis-incompetent cyt c.

## Data Availability

Not applicable.

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
