# Peer review of "Biological Action of Singlet Molecular Oxygen from the Standpoint of Cell Signaling, Injury and Death"

_molecules, 2023, doi:10.3390/molecules28104085_

Round 1

Reviewer 1 Report

This comprehensive review article focuses on the reactivity towards biomolecules of singlet oxygen (1O2) generated either by type II photosensitization reactions or from suitable endoperoxide precursors. In addition relevant information is provided on the biological role of this reactive oxygen species (ROS) in cell signaling as well as its implication in deleterious biological effects and cell death. Altogether the contribution appears to be worth of publication, however after consideration of several remarks that essentially concern chemical aspects as further detailed below.

Major remarks.

1 – Photodynamic effects. Photosensitized reactions are the main contributors of 1O2 generation in biological systems through type II mechanisms involving energy transfer from triplet excited molecules to triplet oxygen. Photosensitizers may also act according to competitive type I photosensitization mechanism. This mostly involves charge transfer between suitable targets (characterized by a low oxidation potential such as guanine or tryptophan) and a photosensitizer in its triplet excited state. The one-electron oxidation of targets gives rise to unstable radical cations that further react after hydration/deprotonation to generate oxidation products. In addition, superoxide anion radical (O2•-) is produced mostly through oxidation of radical anion of photosensitizers by O2 and also to a lesser extent by charge transfer reaction between triplet excited photosensitizers and O2. Low reactive O2•- may dismutase either spontaneously or enzymatically into hydrogen peroxide (H2O2), a precursor of highly reactive hydroxyl radical (OH) through Fenton reaction. Further details on these photosensitized oxidation reactions are provided in recent reviews articles (Baptista et al, Photochem Photobiol 2017, 93, 912-9; ibid, 2021, 97, 1456-83).

2 – Biological oxidation reactions. In addition to the implication of oxidation pathways triggered by the main ROS including OH and 1O2, it is worthwhile to mention one–electron oxidation reactions as other sources of oxidatively generated damage to biomolecules. Thus in addition to type I photosensitized reactions (see previous remark) one may quote direct effect of ionizing radiation (Sharma et al, J Phys Chem B 2011, 115, 4843-55), bi-photonic high intensity UVC laser pulses (Cadet et al, Photochem Photobiol 2019, 95, 59-72) and one-electron agents such as metabolized potassium bromate (Chipman et al, Toxicology 206, 221, 187-9) and carbonate radical anion (Joffe et al, Chem Res Toxicol  2003, 16, 1528-38) that is issued from the decomposition of nitrosoperoxycarbonate, the reaction product of peroxynitrite and bicarbonates.

3 – Use of heat-labile endoperoxides as a clean source of 1O2 for highly specific oxidation of cellular biomolecules. This is a convenient chemical system that has been applied for 1O2 oxidation of the DNA guanine base in cells (Ravanat et al, J Biol Chem 2000, 275, 40601-04; Carcinogenesis 2002, 23, 1911-18).

4 – Cholesterol hydroperoxides as specific biomarkers of type I and II type photosensitization reactions (pg 7). Cholesterol through the measurement of dedicated oxidatively generated hydroperoxides has been shown to be a relevant membrane lipid constituent for assessing the contribution of 1O2-mediated oxidation reactions with respect to type I photosensitized oxidation pathways  (Korytowski & Girotti, Photochem Photobiol 1995, 70, 484-9; Girotti & Korytowski, 2019, 95, 73-82).

5 Oxidation of  nucleotides by 1O2 (pg 9). This section needs substantial amendments (see below).

- As a first remark UVB radiation is not ‘known to cause oxidative DNA damage’ (pg 9, ln 369). In fact UVB is considered to be a very poor DNA oxidizing agent in most living systems. This is the case in mammalian cells where it has been shown that the formation of DNA single-strand breaks or 8-oxo-7,8-dihydroguanine (8-oxoGua), another relevant biomarker of DNA oxidation, represents at best 1% of the overall amount of dimeric pyrimidine photoproducts generated by exposure to UVB radiation (Cadet et al, Photochem Photobiol 2015, 91,140-55; Cadet & Douki, Photochem Photobiol Sci 2018, 17, 1816-41). In contrast UVA that still generate cyclobutane pyrimidine dimers as the main photoproducts appears to be a better oxidizing agent through the predominant photosensitized generation of 1O2 with respect to OH-mediated oxidation (Pouget et al, Chem Res Toxicol 12, 541-9).

It should also be noted that UVB-induced skin cancer is essentially due to the photo-induced formation of cytosine containing bipyrimidine photoproducts and not 8-oxoGua.

 -  Oxidation of guanine by 1O2 does not show a “similar trend to that for the oxidative modification of guanine by OH’ (pg 9, lns 375-376). This is not a correct statement. The only similarity since both reactive oxygen species are able to generate 8-oxo-7,8-dihydroguanine (8-oxoGua), however according to different mechanisms. It should also be noted that the reactivity of OH toward guanine is similar to that for other bases and the 2-deoxyribose moiety. This explains why OH-mediated oxidation reactions with DNA generate a wide diversity of base modification whereas 1O2 is only able to produce 8-oxoGua (Cadet et al Free Radic Biol 2017, 107, 13-34). It may be added that guanine is about one hundred-folds more susceptible to type I photosensitized oxidation reactions than other purine and pyrimidine bases.

- NAD(P)H) as a target for 1O2 oxidation. It should make clear that this reaction is likely to involve the pyridine moiety through a reversible NADH/NAD conversion. However the  adenine moiety that does not react with 1O2 is not implicated in this process (a correction is necessary in the abstract, pg 1, ln 20).

Minor issues

Pg 9, ln 369 and elsewhere: ‘oxidative DNA damage’ should be ‘oxidatively generated DNA damage’ since DNA oxidation products do not show oxidizing features.

Pg 9, ln 372 and elsewhere: ‘guanosine’ that is a RNA component and not a DNA 2’-deoxyribonucleoside should be ‘2’-deoxyguanosine’.

Pg 9, lns 373,374: ‘7,8-dihydro-8-oxo-2’-deoxyguanosine (8-OH-dG)’ should be preferentially called ‘8-oxo-7,8-dihydro-2’-deoxyguanosine (8-oxodG) since it is considred as the hydroxylated derivative of virtual 7,8-dihydroguanine. Furthermore the abbreviation ‘8-oxodG’ is preferable to ‘8-OHdG’ since it was shown that in solution the 6,8-diketo tautomer is predominant over the 8-enol form (Cooke et al, Chem Res Toxicol 2010, 23, 705-7; Cadet et al, Free Radic Res 2012, 46, 367-81).

Author Response

Thank you very much for your efforts on reviewing and criticism on our manuscript. We are grateful to the reviewers for their valuable comments. We understand that there are many excellent reviews in the fields of chemical, biological, and medical aspects regarding singlet oxygen. Since this review focuses on recent findings, notably application of singlet oxygen for researches on basic biology and medicine, the minimum description is kept concerning chemical issues, notably which have been established.

Our individual responses follow after their comments.

This comprehensive review article focuses on the reactivity towards biomolecules of singlet oxygen (1O2) generated either by type II photosensitization reactions or from suitable endoperoxide precursors. In addition relevant information is provided on the biological role of this reactive oxygen species (ROS) in cell signaling as well as its implication in deleterious biological effects and cell death. Altogether the contribution appears to be worth of publication, however after consideration of several remarks that essentially concern chemical aspects as further detailed below.

Major remarks.

1 – Photodynamic effects. Photosensitized reactions are the main contributors of 1O2 generation in biological systems through type II mechanisms involving energy transfer from triplet excited molecules to triplet oxygen. Photosensitizers may also act according to competitive type I photosensitization mechanism. This mostly involves charge transfer between suitable targets (characterized by a low oxidation potential such as guanine or tryptophan) and a photosensitizer in its triplet excited state. The one-electron oxidation of targets gives rise to unstable radical cations that further react after hydration/deprotonation to generate oxidation products. In addition, superoxide anion radical (O2•-) is produced mostly through oxidation of radical anion of photosensitizers by O2 and also to a lesser extent by charge transfer reaction between triplet excited photosensitizers and O2. Low reactive O2•- may dismutase either spontaneously or enzymatically into hydrogen peroxide (H2O2), a precursor of highly reactive hydroxyl radical (OH) through Fenton reaction. Further details on these photosensitized oxidation reactions are provided in recent reviews articles (Baptista et al, Photochem Photobiol 2017, 93, 912-9; ibid, 2021, 97, 1456-83).

Responses: Thank you for your detailed commentary and literature from a chemical point of view.As noted in the aim, this review focuses on the recent advances in biological effects of singlet oxygen and possible application for medicine.Therefore, we limited description on the chemical issue along with literature, which are provided.

2 – Biological oxidation reactions. In addition to the implication of oxidation pathways triggered by the main ROS including OH and 1O2, it is worthwhile to mention one–electron oxidation reactions as other sources of oxidatively generated damage to biomolecules. Thus in addition to type I photosensitized reactions (see previous remark) one may quote direct effect of ionizing radiation (Sharma et al, J Phys Chem B 2011, 115, 4843-55), bi-photonic high intensity UVC laser pulses (Cadet et al, Photochem Photobiol 2019, 95, 59-72) and one-electron agents such as metabolized potassium bromate (Chipman et al, Toxicology 206, 221, 187-9) and carbonate radical anion (Joffe et al, Chem Res Toxicol  2003, 16, 1528-38) that is issued from the decomposition of nitrosoperoxycarbonate, the reaction product of peroxynitrite and bicarbonates.

Responses: Thank you very much for valuable information. However, this review focuses on recent findings, notably application of singlet oxygen for basic biology and medicine,so the issue raised by the reviewer is a bit out of focus. Therefore, we did not mention them any further.

3 – Use of heat-labile endoperoxides as a clean source of 1O2 for highly specific oxidation of cellular biomolecules. This is a convenient chemical system that has been applied for 1O2 oxidation of the DNA guanine base in cells (Ravanat et al, J Biol Chem 2000, 275, 40601-04; Carcinogenesis 2002, 23, 1911-18).

Responses: Thank you very much for valuable information. According to the comments, we have added some statement along with reference, which are provided.

4 – Cholesterol hydroperoxides as specific biomarkers of type I and II type photosensitization reactions (pg 7). Cholesterol through the measurement of dedicated oxidatively generated hydroperoxides has been shown to be a relevant membrane lipid constituent for assessing the contribution of 1O2-mediated oxidation reactions with respect to type I photosensitized oxidation pathways  (Korytowski & Girotti, Photochem Photobiol 1995, 70, 484-9; Girotti & Korytowski, 2019, 95, 73-82).

Responses: Thank you very much for valuable information. According to the comments, we have added some statement and reference.

5 – Oxidation of  nucleotides by 1O2 (pg 9). This section needs substantial amendments (see below).

- As a first remark UVB radiation is not ‘known to cause oxidative DNA damage’ (pg 9, ln 369). In fact UVB is considered to be a very poor DNA oxidizing agent in most living systems. This is the case in mammalian cells where it has been shown that the formation of DNA single-strand breaks or 8-oxo-7,8-dihydroguanine (8-oxoGua), another relevant biomarker of DNA oxidation, represents at best 1% of the overall amount of dimeric pyrimidine photoproducts generated by exposure to UVB radiation (Cadet et al, Photochem Photobiol 2015, 91,140-55; Cadet & Douki, Photochem Photobiol Sci 2018, 17, 1816-41). In contrast UVA that still generate cyclobutane pyrimidine dimers as the main photoproducts appears to be a better oxidizing agent through the predominant photosensitized generation of 1O2 with respect to OH-mediated oxidation (Pouget et al, Chem Res Toxicol 12, 541-9).

It should also be noted that UVB-induced skin cancer is essentially due to the photo-induced formation of cytosine containing bipyrimidine photoproducts and not 8-oxoGua.

Responses: Thank you very much for pointing out. According to the comments, we have added some statement along with reference.

-  Oxidation of guanine by 1O2 does not show a “similar trend to that for the oxidative modification of guanine by OH’ (pg 9, lns 375-376). This is not a correct statement. The only similarity since both reactive oxygen species are able to generate 8-oxo-7,8-dihydroguanine (8-oxoGua), however according to different mechanisms. It should also be noted that the reactivity of OH toward guanine is similar to that for other bases and the 2-deoxyribose moiety. This explains why OH-mediated oxidation reactions with DNA generate a wide diversity of base modification whereas 1O2 is only able to produce 8-oxoGua (Cadet et al Free Radic Biol 2017, 107, 13-34). It may be added that guanine is about one hundred-folds more susceptible to type I photosensitized oxidation reactions than other purine and pyrimidine bases.

Responses: Thank you very much for valuable information. According to the comments, we have corrected the statement.

- NAD(P)H) as a target for 1O2 oxidation. It should make clear that this reaction is likely to involve the pyridine moiety through a reversible NADH/NAD conversion. However the adenine moiety that does not react with 1O2 is not implicated in this process (a correction is necessary in the abstract, pg 1, ln 20).

Responses: Thank you for pointing out. We misread the results of the paper (original ref 90), which shows oxidation of NAD(P)H to NAD(P)+ by singlet oxygen. We have corrected the corresponding statement and simply removed the words from the abstract because they do not fit the sentence.

Minor issues

Pg 9, ln 369 and elsewhere: ‘oxidative DNA damage’ should be ‘oxidatively generated DNA damage’ since DNA oxidation products do not show oxidizing features.

Responses: Thank you very much for kind advice. We have changed ‘oxidative DNA damage’ to ‘oxidatively generated DNA damage’ throughout the text.

Pg 9, ln 372 and elsewhere: ‘guanosine’ that is a RNA component and not a DNA 2’-deoxyribonucleoside should be ‘2’-deoxyguanosine’.

Responses: Thank you very much for kindly pointing out. We have corrected ‘guanosine’ to ‘2’-deoxyguanosine’ throughout the text.

Pg 9, lns 373,374: ‘7,8-dihydro-8-oxo-2’-deoxyguanosine (8-OH-dG)’ should be preferentially called ‘8-oxo-7,8-dihydro-2’-deoxyguanosine (8-oxodG) since it is considered as the hydroxylated derivative of virtual 7,8-dihydroguanine. Furthermore the abbreviation ‘8-oxodG’ is preferable to ‘8-OHdG’ since it was shown that in solution the 6,8-diketo tautomer is predominant over the 8-enol form (Cooke et al, Chem Res Toxicol 2010, 23, 705-7; Cadet et al, Free Radic Res 2012, 46, 367-81).

Responses: Thank you very much for kind advice. We have corrected ‘7,8-dihydro-8-oxo-2’-deoxyguanosine (8-OH-dG)’ to ‘8-oxo-7,8-dihydro-2’-deoxyguanosine (8-oxodG)’ and ‘8-OHdG’ to ‘ ‘8-oxodG’

Reviewer 2 Report

The authors have reviewed new aspects of Singlet Oxygen in the field of photodynamic therapy (PDT).

While there are several reviews on the mechanism and application of PDT, the authors in this review have opened a new discussion on the mechanism of Singlet Oxygen generation towards anti-tumor application.

The title of this review seems well-fitted with the scope of this journal, and it provides a comprehensive and balanced overview of the topic. However, there are some minor and major revisions that should be addressed before being appropriate for publication, as follow;

……………

Figure-1: This figure actually is representing the “Jablonski diagram”. Please mention the name (Jablonski diagram), and properly mention the citation:

Jablonski, A. Efficiency of anti-Stokes fluorescence in dyes. Nature 131, 839-840 (1933).

Figure-3: This figure is well representing the oxidation of biomolecules. Please mention the type of each molecule. It is better to mention these terms for each one in the figure, as A) lipid, B) Protein enzyme, C) Nucleotide.

In section “4. 1. 1O2 in protection against microbial infection”, it is better to discuss “bacterial” and “viral” infections in 2 separate paragraphs.

For bacterial infection; the authors may discuss the cellular structure of negative and positive bacteria (lipid membrane, OM, peptide glycan) how could be affected by singlet oxygen. As an example, this review has discussed the Microbial Biofilm Structural Challenge: https://doi.org/10.3390/ijms23063209

For viral infection, the authors should consider the nature of viral particles (that are not live microorganisms and have a lack of metabolism). This could be interesting to discuss the singlet oxygen effect on both virion and virus-infected cell, as studies here;

Sadraeian, M. et al. Photoinduced Photosensitizer–Antibody Conjugates Kill HIV Env-Expressing Cells, Also Inactivating HIV. ACS omega 6, 16524-16534 (2021).

Figure-5: This figure looks like a key figure concluding this review. However, the quality of this figure is low. It is too colorful, with inappropriate colors. To improve it, please

-          Show the apoptosis with the same color for both parts A and B.

-          Show the cell membrane in black

-          Show the arrows in black

-          Make the mitochondria more visible to be easily recognizable at the first glance

-          In part B, please mention the name of procaspases (I guess there are Procaspases 3 and 9)

No issues were detected.

Author Response

Thank you very much for your efforts on reviewing and criticism on our manuscript. We are grateful to the reviewers for their valuable comments. We understand that there are many excellent reviews in the fields of chemical, biological, and medical aspects regarding singlet oxygen. Since this review focuses on recent findings, notably application of singlet oxygen for researches on basic biology and medicine, the minimum description is kept concerning chemical issues, notably which have been established.

Our individual responses follow after their comments.

The authors have reviewed new aspects of Singlet Oxygen in the field of photodynamic therapy (PDT).

While there are several reviews on the mechanism and application of PDT, the authors in this review have opened a new discussion on the mechanism of Singlet Oxygen generation towards anti-tumor application.

The title of this review seems well-fitted with the scope of this journal, and it provides a comprehensive and balanced overview of the topic. However, there are some minor and major revisions that should be addressed before being appropriate for publication, as follow;

……………

Figure-1: This figure actually is representing the “Jablonski diagram”. Please mention the name (Jablonski diagram), and properly mention the citation:

Jablonski, A. Efficiency of anti-Stokes fluorescence in dyes. Nature 131, 839-840 (1933).

 Responses: Thank you very much for kind advice and providing the reference. We have mentioned the “Jablonski diagram” and cited the reference according to advice.

Figure-3: This figure is well representing the oxidation of biomolecules. Please mention the type of each molecule. It is better to mention these terms for each one in the figure, as A) lipid, B) Protein enzyme, C) Nucleotide.

Responses: Thank you very much for kind advice. Although each molecules are mentioned under (A) P-PUF (arachidonate), (B) Enzymes (catalytic Cys) and (C) Nucleotide (guanine) in or original figure, we have added larger classification terms according to the advice. 

In section “4. 1. 1O2 in protection against microbial infection”, it is better to discuss “bacterial” and “viral” infections in 2 separate paragraphs.

For bacterial infection; the authors may discuss the cellular structure of negative and positive bacteria (lipid membrane, OM, peptide glycan) how could be affected by singlet oxygen. As an example, this review has discussed the Microbial Biofilm Structural Challenge: https://doi.org/10.3390/ijms23063209

For viral infection, the authors should consider the nature of viral particles (that are not live microorganisms and have a lack of metabolism). This could be interesting to discuss the singlet oxygen effect on both virion and virus-infected cell, as studies here;

Sadraeian, M. et al. Photoinduced Photosensitizer–Antibody Conjugates Kill HIV Env-Expressing Cells, Also Inactivating HIV. ACS omega 6, 16524-16534 (2021).

Responses: Thank you very much for kind advice. We emphasized mechanism of anti-microbial and anti-tumor effects of singlet oxygen from aspects of oxidation of biological molecules. Because proteins, lipids, and nucleic acids are common targets in any organisms, so we discussed about singlet oxygen-mediated oxidation of them, instead of discussing microbial or cancer cells individually. Thus, we consider the subjects that are proposed by the reviewer is out of focus of this manuscript. In addition, as the reviewer mentioned, the issue has been extensively overviewed in superior review articles.

Figure-5: This figure looks like a key figure concluding this review. However, the quality of this figure is low. It is too colorful, with inappropriate colors. To improve it, please

Responses: We thank for your comments on the figure. Our responses individually followed your comments.

-          Show the apoptosis with the same color for both parts A and B.

Responses: Since apoptosis does not progress in cell, at high levels of singlet oxygen (B), the color is intentionally lightened and the arrows are dashed. We believe that the current situation makes it easier for readers to understand the intention.

-          Show the cell membrane in black

Responses: We used different color for Fig (A) and (B) because the cell in (A) dies by apoptosis but that in (B) dies by ferroptosis. However, as the reviewer pointed, use of multiple color makes it difficult to understand. So we have changed the color of the cell membrane in grey according to the comment.

-          Show the arrows in black

Responses: We used different color and thickness for arrows intentionally as above. Black is too strong, so we have changed them to grey. Because arrows from singlet oxygen indicate effects, which is main signaling pathway of this figure, we have not changed.  

-          Make the mitochondria more visible to be easily recognizable at the first glance

Responses: Because mitochondria is not main component in this figure, we made it faint. However it was too faint, so we have thickened it enough not to disturb the signal flow.

-          In part B, please mention the name of procaspases (I guess there are Procaspases 3 and 9)

Responses: Thank you for asking. We intentionally omitted their names to avoid complexity of the figure because they are the same as Fig (A). Now we have added the same name according to your comments. Procaspases are not converted to mature caspases due to oxidative inactivation, so we did not added names mature caspase 3 or 9. 

Round 2

Reviewer 1 Report

The authors have made appropriate changes in the text by adequately addressing all the reviewer's requests. Therefore the revised version of the manuscript is recommended for publication pending however a few minor corrections:

Figure 1 and elsewhere: 'O2-' should be ' O2•-

Figure 3: ‘8-oxo guanine’ should be ‘8-oxo-7,8-dihydroguanine’

Legend of  Figure 3: ‘8-Oxoguanine’ should be ‘8-Oxo-7,8-dihydroguanine’